# Physalin A, 13,14-Seco-16, 24-Cyclo-Steroid, Inhibits Stemness of Breast Cancer Cells by Regulation of Hedgehog Signaling Pathway and Yes-Associated Protein 1 (YAP1)

**DOI:** 10.3390/ijms22168718

**Published:** 2021-08-13

**Authors:** Yu-Chan Ko, Hack Sun Choi, Ren Liu, Dong-Sun Lee

**Affiliations:** 1Interdisciplinary Graduate Program in Advanced Convergence Technology and Science, Jeju National University, Jeju 63243, Korea; koyuchan94@gmail.com (Y.-C.K.); liuren0308@gmail.com (R.L.); 2Subtropical/Tropical Organism Gene Bank, Jeju National University, Jeju 63243, Korea; choix074@jejunu.ac.kr; 3Bio-Health Materials Core-Facility Center, Jeju National University, Jeju 63243, Korea; 4Practical Translational Research Center, Jeju National University, Jeju 63243, Korea; 5Faculty of Biotechnology, College of Applied Life Sciences, Jeju National University, Jeju 63243, Korea

**Keywords:** breast cancer stem cells (BCSCs), physalin A, Hedgehog signaling pathway, GLI1, Hippo signaling pathway, YAP1, mammospheres

## Abstract

The Hedgehog (HH) signaling pathway plays an important role in embryonic development and adult organ homeostasis. Aberrant activity of the Hedgehog signaling pathway induces many developmental disorders and cancers. Recent studies have investigated the relationship of this pathway with various cancers. GPCR-like protein Smoothened (SMO) and the glioma-associated oncogene (GLI1) are the main effectors of Hedgehog signaling. Physalin A, a bioactive substance derived from *Physalis alkekengi*, inhibits proliferation and migration of breast cancer cells and mammospheres formation. Physalin A-induced apoptosis and growth inhibition of mammospheres, and reduced transcripts of cancer stem cell (CSC) marker genes. Physalin A reduced protein expressions of SMO and GLI1/2. Down-regulation of SMO and GLI1 using siRNA inhibited mammosphere formation. Physalin A reduced mammosphere formation by reducing *GLI1* gene expression. Down-regulation of GLI1 reduced CSC marker genes. Physalin A reduced protein level of YAP1. Down-regulation of YAP1 using siRNA inhibited mammosphere formation. Physalin A reduced mammosphere formation through reduction of *YAP1* gene expression. Down-regulation of YAP1 reduced CSC marker genes. We showed that treatment of MDA-MB-231 breast cancer cells with GLI1 siRNA induced inhibition of mammosphere formation and down-regulation of YAP1, a Hippo pathway effector. These results show that Hippo signaling is regulated by the Hedgehog signaling pathway. Physalin A also inhibits the canonical Hedgehog and Hippo signaling pathways, CSC-specific genes, and the formation of mammospheres. These findings suggest that physalin A is a potential therapeutic agent for targeting CSCs.

## 1. Introduction

Breast cancer (BC) is invasive cancer and involves different areas of the breast (lobules, ducts, and connective tissue) and shows different clinical outcomes. Based on the cancer response, BC can be divided into estrogen receptor(ER)-positive, progesterone receptor (PR)-positive, ER/PR-positive, human epidermal growth factor receptor-2 (HER2)-positive, and triple-negative breast cancer (TNBC) [1]. Breast cancer stem cells (BCSCs) are a small population of BC cells that play a critical role in the metastasis of BC to other organs and are a leading cause of tumor progression and resistance against conventional therapy. Therefore, targeting BCSCs may be a potential approach for the treatment of BC. Triple-negative breast cancer (TNBC) is a heterogeneous group of tumors that exhibit diverse histological characteristics, molecular profile, and therapeutic response. TNBCs are characterized by a lack of amplification of the expressions of estrogen receptor, progesterone receptor, and human epidermal growth factor receptor 2 (HER2) [1]. Approximately 10–15% of all breast cancers are TNBC [2]. Compared to other breast cancer types, TNBC is associated with poor prognosis and occurs at a higher frequency in younger women [3]. Breast cancer stem cells (BCSCs) are heterogeneous subpopulations of breast cancer cells that have properties of differentiation and self-renewal [4]. Additionally, these BCSC subpopulations are known to influence therapeutic response and clinical outcomes [5]. Owing to these characteristics, the development of treatments targeting BCSCs is an emerging research hotspot. However, there are no standardized strategies for TNBCs and BCSCs [6].

Physalin A is an active withanolide derived from *Physalis alkekengi* var. *franchetii* (Solanaceae) which is named “Jin-deng-long” in China [7]. Traditional Chinese medicine using the calyx of this plant is commonly adopted for the treatment of excessive phlegm, pharyngitis, phemphigus, cough, sore throat, hepatitis, eczema, dysuria, and tumors [8]. These data suggest that the genus *Physalis* is an important source of withanolide-type molecules with potential therapeutic effect against human diseases [9]. Studies have demonstrated anti-tumor and anti-inflammatory properties of Physalin A. Nrf2 pathway was regulated by physalin A through ERK and p38 for induction of detoxifying enzymes [10]. Physalin A was shown to induce apoptosis and protective autophagy in HT1080 human fibrosarcoma cells [7]. Physalin A attenuated inflammation by down-regulating c-Jun NH2 kinase (JNK) phosphorylation/Activator Protein 1 (AP-1) activation and up-regulating the anti-oxidative activity [11]. In addition, degradation of nuclear factor kappa B alpha (IκBα) and nuclear translocation of nuclear factor-κB (NF-κB) was blocked by physalin A, which induced an anti-inflammatory effect [12]. In human non-small cell lung cancer cells, physalin A-induced G2/M phase cell cycle arrest and regulated the p38 MAPK/ROS pathway [13]. In another study, physalin A was shown to induce apoptosis of prostate cancer cells by regulation of JNK and activation of ERK [14]. Compared with 5-fluorouracil or paclitaxel, physalin A showed no inhibitory effect on human peripheral blood mononuclear cells and exhibited lower toxicity on normal human cells [7]. Thus, physalin A can be considered as a potential therapeutic agent for clinical use.

Hedgehog (HH) signaling pathway plays a fundamental role in normal embryonic development and postnatal tissue regeneration as well as in organogenesis and homeostasis of almost all organs [15,16,17]. Its biological influence is mediated through a signaling cascade that alters the balance between the activator and repressor forms of glioma-associated oncogene (GLI) transcription factors [16]. The components involved in Hedgehog signaling include Hedgehog ligands (Sonic Hedgehog, Indian Hedgehog, and Desert Hedgehog), the seven-span transmembrane G protein-coupled receptor (GPCR)-like protein Smoothened (SMO), the twelve-span transmembrane canonical receptor protein Patched (PTCH1), and the glioma-associated oncogene (GLI1, GLI2, and GLI3) proteins [18]. Aberrant activity of the Hedgehog signaling pathway has been implicated in several developmental disorders and cancers [19]. Constitutive pathway activation resulting from mutations of Hedgehog pathway components is frequently observed in some cancers [20,21]. In cancer stem cells, Hedgehog signaling is known to drive the CSC phenotype through the regulation of stemness-determining genes such as *Nanog, Oct4*, and *Sox2* [22,23,24,25,26] and regulate self-renewal and differentiation of CSCs [27]. These findings indicate that characterization of the molecular mechanisms of the Hedgehog signaling pathway is essential for developing strategies for the prevention and treatment of cancers or CSCs.

The Hippo signaling pathway, a serine/threonine kinase cascade, is related to the proliferation of embryonic and somatic stem cells. Previous studies have shown that the Hippo signaling pathway is a tumor suppressor pathway that induces apoptosis and inhibits proliferation [28]. In particular, YAP and TAZ, which are the main effectors of the Hippo pathway, are essential for the maintenance of BCSCs. In breast cancer cells, TAZ-TEAD-Cyr61/CTGF, the Hippo signaling pathway is an important modifier of the paclitaxel response [29]. TAZ is required for the metastatic activity, self-renewal ability, tumor-initiating capacity, and chemoresistance of BCSCs [30]. YAP-induced stemness in mammary epithelial cells and breast cancer is mediated by SRF-IL6 [31]. These findings indicate that the Hippo signaling pathway is determinant for treating BCSCs. However, there has been no report on anti-CSCs activity and molecular mechanisms of physalin A from *Physalis alkekengi* var. *franchetii*. In the present study, we studied physalin A as a mammosphere formation inhibitor against BCSCs which suppresses mammosphere formation in breast cancer cell lines. We investigated whether physalin A inhibits the formation of BCSCs through the regulation of the Hedgehog signaling pathway and yes-associated protein 1 (YAP1).

## 2. Results

### 2.1. Physalin A Inhibits the Proliferation of MDA-MB-231, MDA-MB-453, HCC-1937, and MCF-7 Breast Cancer Cells and Mammosphere Formation

TNBC cell lines (MDA-MB-231, MDA-MB-453, and HCC-1937) and non-TNBC cell lines (MCF-7) were cultured in 96-well plates. The effect of physalin A (Figure 1B and Appendix A) on cell viability was assessed using MTS assay. As shown in Figure 1B and Appendix A, physalin A suppressed the proliferation of breast cancer cell lines in a dose-dependent manner. The migration and colony formation of MDA-MB-231 and MCF-7 cells were inhibited by physalin A (Figure 1C,D). To assess whether physalin A has an inhibitory effect on mammosphere formation, we cultured MDA-MB-231 (1 × 10^4^ cells/well), MDA-MB-453 (2 × 10^4^ cells/well), HCC-1937 (4 × 10^4^ cells/well), and MCF-7 (4 × 10^4^ cells/well) in ultralow attachment plates with/without physalin A (10 μM). Physalin A decreased the number and size of the mammospheres (Figure 1E and Appendix A).

### 2.2. Physalin A Decreases the CD44^high^/CD24^low^ and ALDH1-Expressing Subpopulations

CD44^high^/CD24^low^ and ALDH1-expressing subpopulations are BCSC populations. We cultured MDA-MB-231 cells in 6-well culture plates and then treated with/without physalin A for 24 h. Physalin A reduced the population of CD44^high^/CD24^low^-expressing subpopulation of MDA-MB-231 breast cancer cells from 75.7% to 42.4% (Figure 2A). Similarly, the ALDH assay showed that physalin A decreased the ALDH1-positive subpopulation of MDA-MB-231 cells from 5.3% to 1.3% (Figure 2B). These results indicate that physalin A reduced the frequency of BCSC populations, CD44^high^/CD24^low^, and ALDH1-expressing subpopulations.

### 2.3. Physalin A Induces BCSC Apoptosis and Inhibits CSC-Specific Gene Transcription and Growth of Mammospheres

We examined the effects of physalin A on the apoptosis of BCSCs. Early apoptosis (Q2) was induced from 10.5% to 12.8% and late apoptosis (Q1) was induced from 9.8% to 27.4% (Figure 3A). Moreover, physalin A inhibited the expression of CSC-specific genes; *Oct4, CD44, Sox2, c-myc*, and *Nanog* (Figure 3B). To measure the inhibitory effect of physalin A on mammosphere proliferation, MDA-MB-231 mammospheres were treated with physalin A. With the same number of cells, we incubated the single cell from mammospheres in 6 cm diameter dishes for 1, 2, and 3 days. As shown in Figure 3C, physalin A inhibited the proliferation of mammospheres.

### 2.4. Physalin A Regulates the Canonical Hedgehog Signaling Pathway

To determine the cellular mechanism by which physalin A inhibits mammosphere formation, we first checked the Hedgehog signaling pathway. Physalin A reduced the expression of SMO protein (the upper signal effector of the Hedgehog signaling) in mammospheres (Figure 4A and Appendix A). We further confirmed the relationship between SMO and the formation of CSCs. After treating siRNA of SMO, we scanned the plate and measured the MFE. The number of mammospheres was modestly decreased, and the size of mammospheres was also decreased, as shown in Figure 4B. Physalin A also reduced the GLI1 and GLI2 (Figure 4C and Appendix A). siRNA-mediated silencing of GLI1 in MDA-MB-231 cells led to a significant decrease in mammosphere formation. The silencing of GLI1 reduced the number as well as the size of mammospheres (Figure 4D). In BCSCs, physalin A also decreased the transcriptional level of GLI1 (Figure 4E). Silencing of GLI1 also reduced the transcriptional levels of CSC marker genes, *Oct4, Nanog*, and *Sox2*, but not that of *c-myc* (Figure 4F). These results showed that physalin A inhibits the canonical Hedgehog signaling pathway in MDA-MB-231 breast cancer stem cells.

### 2.5. Physalin A Inhibits the Hippo Signaling Pathway Which Is Regulated by GLI1

To study the downstream effects of the Hedgehog signaling pathway, we examined the effect of physalin A on the expression level of YAP1 (an effector of the Hippo signaling pathway) in the mammosphere. Physalin A reduced the total levels of YAP1 protein and also reduced cytosol and nuclear levels of YAP1 in MDA-MB-231 and MCF-7 mammospheres (Figure 5A,B and Appendix A). siRNA-mediated silencing of YAP1 significantly decreased mammosphere formation of MDA-MB-231 cells. The silencing of YAP1 reduced the number as well as the size of mammospheres (Figure 5C). Physalin A induced a decrease in transcripts level of YAP1 in mammospheres (Figure 5D). The expressions of cancer stem cell-related genes, *Sox2* and *Oct4,* were decreased by siRNA of YAP1 (Figure 5E). To examine the crosstalk of YAP1 and GLI1, we examined the level of YAP1 protein in cancer cells after the silencing of GLI1. Down-regulation of GLI1 using siRNA induced a decrease in transcripts and protein level of YAP1 (Figure 6A,B) and reduced cytosolic and nuclear fractions of YAP1 (Figure 6C). YAP1 is essential for the formation of mammospheres. According to available TCGA data, GLI1 is related to YAP1 in breast cancer patients (Figure 7A). In conclusion, our data showed that GLI1 regulates the protein and transcription level of YAP1, and physalin A affects the Hippo signaling pathway via regulation of the Hedgehog signaling pathway. A schematic illustration of the proposed mechanism is presented in Figure 7B.

## 3. Discussion

Breast cancer is the most malignant cancer in women [32]. Triple-negative breast cancer (TNBC) lacks estrogen receptor, progesterone receptor, and HER2 expression and TNBC patients have a higher rate of relapse and a poorer prognosis than other breast cancer patients [3]. Triple-negative breast cancer (TNBC) is typically challenging to treat [33]. TNBC is associated with chemotherapeutic failure and poor prognosis [34,35]. BCSCs are a key determinant of tumor heterogeneity and contribute to the induction of chemoresistance and metastasis [36]. Additionally, BCSCs are responsible for cancer progression, recurrence, and therapeutic resistance [37]. CD44^high^/CD24^low^ and ALDH1 expressions, which are biomarkers of BCSCs, can be regulated during CSC progression [38]. An increasing body of evidence suggests that targeting breast CSCs is a viable therapeutic strategy for breast cancer.

This study aimed to understand the mechanism of action of physalin A derived from the natural product in BCSCs. There is a growing interest in identifying compounds with anticancer potential from herbs and many natural products that are clinically used for chemotherapies. For example, vincristine, irinotecan, etoposide, and paclitaxel are representative compounds derived from plants, which are used in cancer therapy [39]. Physalins, a type of steroids are the characteristic constituents of *Physalis alkekengi L Physalis* are traditionally used in China for the treatment and prevention of tumors.

Physalin A is a steroidal compound. Steroids are widely used in cancer therapy and have been known to possess an anti-cancer effect. For instance, Ciclesonide, a kind of glucocorticoid, inhibits the formation of BCSCs via the GR/YAP pathway [40] and the lung cancer stem cells via the Hedgehog/Sox2 signaling pathway [41]. A steroidal lactone, withaferin A, induces Par-4-dependent apoptosis in prostate cancer cells [42]. In our study, physalin A inhibited the proliferation of breast cancer cells and the formation of mammospheres (Figure 1). Physalin A also induced apoptosis of BCSCs (Figure 3). Physalin A decreased the BCSC biomarkers (such as CD44^high^/CD24^low^ and ALDH1) and reduced the transcriptional levels of CSC-specific genes such as *Oct4, CD44, Sox2, c-myc*, and *Nanog* (Figure 2 and Figure 3). Our data show the inhibitory effect of physalin A on BCSCs.

Hedgehog signaling pathway effectors such as SMO and GLI, are essential for cancer progression. Inhibition of Hedgehog signaling enhanced the delivery of chemotherapy in a mouse model of pancreatic cancer [43]. Inhibition of sonic Hedgehog ligand activity inhibits the growth of small cell lung cancer lines expressing Shh and GLI, but not that of non-small cell lung cancer lacking expression of both Shh and GLI [44]. NF-κB up-regulation is responsible for Shh overexpression which induces breast carcinogenesis [45]. The Hedgehog signaling pathway is a critical mechanism in breast cancer cells [46,47,48,49]. Additionally, Hedgehog signaling and Bmi-1 regulate the self-renewal of BCSCs [50]. An agonist of the A3 adenosine receptor was shown to inhibit the survival of BCSCs via the GLI1/ERK pathway [51]. In our study, physalin A inhibited the formation of BCSCs through down-regulating canonical Hedgehog signaling effectors such as SMO and GLI1. Silencing of SMO and GLI1 using specific siRNAs reduced the size as well as the number of mammospheres. Moreover, GLI1 knockdown decreased the mRNA levels of CSC marker genes, *Oct4, Nanog*, and *Sox2* (Figure 4). These findings indicate that the main effectors of the Hedgehog signaling pathway play a vital role in the maintenance of BCSCs. Some studies have demonstrated the crosstalk of the Hedgehog signaling pathway with various oncogenic pathways. In melanomas, the Hedgehog signaling is regulated by the interaction between GLI1 and RAS-MEK/AKT pathway [52]. PI3K/AKT plays an important role in Hedgehog-dependent tumors [53]. Targeting the crosstalk between major oncogenic signaling pathways is an essential step to improve the anticancer therapeutic efficacy [54]. Hedgehog is related to *CYR61* which is the target gene of the YAP/TAZ complex [55]. As shown in Figure 5 and Figure 6, we examined the crosstalk between Hedgehog signaling and Hippo signaling. GLI1 down-regulation using siRNA reduced the mRNA and protein expressions of YAP1. Physalin A also reduced the level of YAP1. Knockdown of YAP1 affected the expressions of CSC-related genes (Sox2 and Oct4) and the formation of BCSCs. TAZ/YAP activity is also important for the maintenance of BCSCs. TAZ is required for metastasis and chemoresistance of BCSCs [30]. YAP1 protein was shown to be associated with outcomes in patients with luminal A breast cancer [56]. These findings indicate that a better understanding of the crosstalk between these pathways may be shown as potential therapeutic strategies against BCSCs.

Physalin A can inhibit BCSCs via regulation of mammosphere formation and growth, colony formation, and cell migration. Physalin A also inhibits the Hedgehog and Hippo signaling pathway. Our data showed that GLI1 regulates the protein and mRNA expressions of YAP1. These findings suggest that BCSCs are inhibited via inhibition of SMO/GLI1/YAP1 signaling. Our data showed that physalin A is a potential agent for BCSC therapy.

## 4. Materials and Methods

### 4.1. Cell and Mammosphere Culture

TNBC cells (MDA-MB-231, MDA-MB-453, and HCC-1937) and non-TNBC cells (MCF-7) were obtained from the American Type Culture Collection (Rockville, MD, USA) were cultured in Dulbecco’s modified Eagle’s medium (Gibco, Thermo Fisher Scientific, Waltham, MA, USA) supplemented with 10% fetal bovine serum (Gibco, Thermo Fisher Scientific) and 1% penicillin/streptomycin (Gibco, Thermo Fisher Scientific). The four breast cancer cell lines, MDA-MB-231 (1 × 10^4^), MDA-MB-453 (2 × 10^4^), HCC-1937 (4 × 10^4^) and MCF-7 (4 × 10^4^) were cultured in ultralow attachment 6-well plates with MammoCult™ culture medium (STEMCELL Technologies, Vancouver, BC, Canada) supplemented with 4 μg/mL heparin, 0.48 μg/mL hydrocortisone, 100 U/mL penicillin, and 100 μg/mL streptomycin. All cells were incubated in a humidified 5% CO_2_ incubator at 37 °C. After one week of culture, the formation of mammospheres was assessed by obtaining images using a scanner (Epson Perfection V700 PHOTO, Epson, Tokyo, Japan). For counting, regions of interest (ROIs) were created by choosing the desired number of rows and columns (e.g., 2 × 3 for a 6-well plate), and individual ROIs were defined by moving and resizing the provided ROI shapes after selecting the elliptical setting in the NICE program. The background signal of the images was negated using thresholding algorithms, and the selected images were automatically counted. Mammospheres were counted using the NIST’s integrated colony enumerator (NICE) program and assessed as mammosphere formation efficiency (MFE, % of control) [57].

### 4.2. Antibodies and Small Interfering RNAs (siRNAs)

Anti-GLI1 and anti-GLI2 antibodies were purchased from Cell Signaling Technology (Danvers, MA, USA). Anti-β-actin and anti-Lamin B antibodies were purchased from Santa Cruz Biotechnology (Dallas, TX, USA). Anti-SMO antibody was purchased from Bioss (Woburn, MA, USA). Anti-YAP1 antibody was obtained from FineTest^®^ (Wuhan, Hubei, China). Human SMO-, GLI1-, and YAP1-specific siRNAs were purchased from Bioneer (Daejeon, Korea).

### 4.3. Cell Proliferation Assay

MDA-MB-231, MDA-MB-453, HCC-1937, and MCF-7 cells were cultured in a 96-well plate for 24 h and treated with various concentrations (0, 10, 20, 40, 60, 80, and 100 µM) of physalin A (ChemFaces Co., Hubei, China) for 1 day in cell culture medium. Then, cell proliferation assay was performed using CellTiter 96^®^ Aqueous One Solution cell kit (Promega, Madison, WI, USA). We mixed DMEM and aqueous one solution in the ratio of five to one added 100 µL of the mixture to each well and incubated the plate in a 5% CO_2_ incubator at 37 °C for 2 h. The OD_490_ was measured using a Versa Max ELISA microplate reader (Molecular Devices, San Jose, CA, USA).

### 4.4. Colony Formation Assay

For colony formation assay, the two breast cancer cell lines were cultured in 6-well plates at a density of 2 × 10^3^/well and treated with/without 40 µM physalin A at 37 °C for 1 week. The colonies were washed three times with 1× phosphate-buffered saline (PBS), fixed for 10 min using 3.9% formaldehyde, and stained for 1 h with 0.04% crystal violet. The colonies were washed twice with distilled water and dried. The images were acquired by a scanner (Epson). The number of colonies was counted with the NICE software program [58].

### 4.5. Migration Assay

MDA-MB-231 and MCF-7 cells with 2 × 10^6^ cells/plate were cultured in a 6-well plate. After 1 day, the cells were scratched using a microtip. The cells were washed twice with 1× PBS and treated with/without 40 µM physalin A. The plate was placed in the Lionheart^TM^, previously set to 37 °C and 5% CO_2_. First, we selected the scratched areas for obtaining images. Then, imaging of these areas was performed every 30 min over a 12 h period using 4× objective. High contrast bright-field images of the migrated areas were captured using a microscope (Lionheart, Biotek, Winooski, VT, USA) at the Jeju Center of Korea Basic Science Institute (KBSI, core-facility center).

### 4.6. Flow Cytometry Analysis and Aldehyde Dehydrogenase (ALDH1) Activity Assay

MDA-MB-231 cells (1 × 10^6^ cells) were seeded in a 6-well plate for 24 h and treated with DMSO as the control or 20 µM physalin A for 24 h. After treatment with/without 20 µM physalin A for 1 day, cancer cells were detached by using 1× trypsin/EDTA. The detached cells were washed with 1× activated cell sorting (FACS) buffer. A total of 1 × 10^6^ cells were suspended with 100 µL of 1× FACS buffer and 10 µL of FITC-conjugated anti-human CD44 and phycoerythrin (PE)-conjugated anti-human CD24 (BD, San Jose, CA, USA) were added to each sample. After incubating on ice for 20 min, the samples were washed twice with 1× FACS buffer, and then analyzed using an Accuri C6 flow cytometer (BD, San Jose, CA, USA). The ALDH1 assay was performed using an ALDEFLUOR^TM^ assay kit (STEMCELL Technologies). The active reagent boron-dipyrromethene (BODIPY)-aminoacetaldehyde was added to each sample, which converts the reagent to fluorescent BODIPY-aminoacetate via ALDH. The ALDH inhibitor diethylaminobenzaldehyde (DEAB) was used as a negative control. MDA-MB-231 cells were treated with/without 20 µM physalin A for 24 h, and then trypsinized for detachment from the plate. After washing with ALDEFLUOR^TM^ assay buffer as per the manufacturer’s instructions, the proportion of ALDH1-positive cells was assayed using an Accuri C6 flow cytometer [59].

### 4.7. Annexin V/PI Assay and Analysis of CSC Apoptosis

Mammospheres of MDA-MB-231 cancer cells were cultured in an ultralow attachment 6-well plate for 5 days and treated with/without 10 µM physalin A for 2 days. Subsequently, the cells were harvested, dissociated, and incubated with FITC-Annexin V/PI for 30 min at 4 °C. Apoptotic cells were analyzed by the FITC-Annexin V/PI staining method according to the manufacturer’s instructions (BD). The stained samples were assayed using a flow cytometer (Accuri C6).

### 4.8. Gene Expression Analysis

Total RNA was isolated from MDA-MB-231 mammospheres. RT-qPCR was performed using a TOPreal^TM^ One-Step RT-qPCR kit (SYBR Green with low ROX) (Enzynomics, Daejeon, Korea) according to the manufacturer’s instructions. We prepared RT-qPCR mixture containing TOPreal^TM^ One-step RT-qPCR Enzyme MIX 1 µL, 2× TOPreal^TM^ One-step RT-qPCR Reaction MIX (with low ROX) 10 µL, RNA template (100 ng/µL) 2 µL, specific primers-F (10 pmol/µL) 2 µL, specific primers-R (10 pmol/µL) 2 µL, and RNase-free sterile water 3 µL in each sample. The relative transcript expression levels of the target genes were analyzed using the comparative CT method [60]. The specific primers used are listed in Appendix A. The β-actin gene was used as an internal control.

### 4.9. Immunoblot Analysis

Protein extracts of cancer cells treated with/without physalin A were isolated from MDA-MB-231 and MCF-7 breast cancer cells and mammospheres. Mammospheres of MDA-MB-231 and MCF-7 cells were cultured in an ultralow attachment 6-well plate for 5 days and treated with physalin A for 2 days. After washing twice with 1× PBS, the cell pellets were resuspended in lysis buffer. For extraction of total protein, the cell pellets were suspended with radio-immunoprecipitation assay (RIPA) buffer (Thermo Fisher Scientific, Rockford, IL, USA) supplemented with 10 mM protease inhibitor, 10 mM sodium fluoride (NaF), and 10 mM sodium vanadate. The samples were incubated on ice for 30 min and then micro-centrifuged at 14,000× *g* for 15 min. The resulting supernatant contained the total proteins. For extraction of cytosolic and nucleic protein, we followed the previously described method [58]. The washed cell pellets were resuspended in buffer A (pH 7.9 of 10 mM HEPES, 1.5 mM MgCl_2_, 10 mM KCl, 0.05% NP-40, 0.5 mM DTT, 10 mM protease inhibitor, 10 mM NaF, and 10 mM sodium vanadate) and the lysates were micro-centrifuged at 10,000× *g* for 5 min. The supernatant contains the cytosolic protein and the resulting pellet contains the nucleus. The nuclear pellet was dissolved with RIPA buffer with 10 mM protease inhibitor, 10 mM NaF, and 10 mM sodium vanadate. The samples were placed on ice for 30 min and then micro-centrifuged at 14,000× *g* for 15 min. The resulting supernatant contains the nucleic protein. The proteins were separated using 8% or 10% sodium dodecyl sulfate-polyacrylamide gel electrophoresis (SDS-PAGE) gel, which was conducted using a tris-glycine buffer. Separated proteins were transferred to polyvinylidene fluoride (PVDF) membranes (Millipore, Burlington, MA, USA). The membranes were incubated with Odyssey blocking buffer at room temperature for 1 h and then incubated overnight with primary antibodies at 4 °C. On the next day, the PVDF membranes were washed thrice with PBS-Tween 20 (0.1%, *v/v*), and incubated with IRDye 680RD- and IRDye 800W-conjugated secondary antibodies for 1 h at room temperature. The signals were detected using an Odyssey CLx imaging system (LI-COR, Lincoln, NE, USA).

### 4.10. Small Interfering RNA (siRNA)

MDA-MB-231 cells were seeded in a 6-well plate at a density of 1.0 × 10^6^ cells/plate. To examine the effect of SMO, GLI1, and YAP1 on mammosphere formation, the cells were transfected with siRNAs targeting the human *SMO*, *GLI1*, and *YAP1* genes. The SMO siRNA (NM_005631.4), GLI1 siRNA (NM_005269.2), YAP1 siRNA (NM_006106.4), and a scrambled siRNA were obtained from Bioneer. Lipofectamine^®^ 3000 (Thermo Scientific) was used to transfect siRNAs according to the manufacturer’s instructions. MDA-MB-231 cells were cultured until the achievement of 70% confluence. We diluted 4 μL of Lipofectamine^®^ 3000 reagent in 125 μL of Opti-Minimal Essential Medium (MEM)^®^ medium and prepared a master mix of siRNAs by diluting 5 μg of siRNAs in 125 μL of Opti-MEM^®^ medium in each tube. Subsequently, we mixed the diluted Lipofectamine^®^ 3000 reagent and diluted siRNAs (for control, only Opti-MEM^®^ and diluted Lipofectamine^®^ 3000 were mixed with a scrambled siRNA) and incubated it for 15 min at room temperature. The siRNA-lipid complex was added to each well and the cells were incubated for 2–4 days at 37 °C. The protein expressions of SMO, GLI1, and YAP1 were determined by Western blot analysis.

### 4.11. Statistical Analysis

All variables are expressed as mean ± standard deviation (SD) values from three independent experiments. One-way ANOVA was used for statistical analysis. Data analysis was performed using GraphPad Prism 8.0 software (GraphPad Software Inc., San Diego, CA, USA).

## 5. Conclusions

In this study, physalin A inhibited the formation of BCSCs and decreased the transcript levels of BCSC-related genes (*Oct4, CD44, Sox2, c-myc,* and *Nanog*). Physalin A also reduced the CD44^high^/CD24^low^ and ALDH1-expressing subpopulations. In BCSCs, this compound was found to inhibit the Hedgehog signaling pathway, especially the important effectors such as SMO and GLI. The significant effector of the Hippo pathway, YAP, was decreased by physalin A and siRNA of GLI1. Our findings suggest that physalin A regulates the Hedgehog/Hippo signaling pathway and can be a potential agent targeting BCSCs.

## Figures and Tables

**Figure 1 ijms-22-08718-f001:**
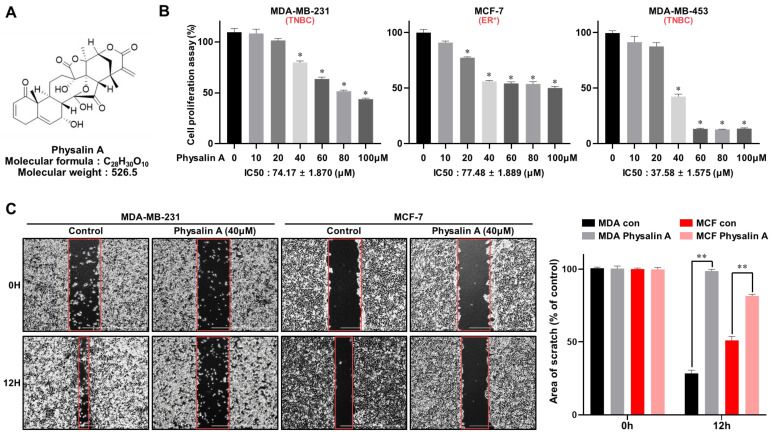
Physalin A inhibits breast cancer cell viability and mammosphere formation efficiency. (**A**) Molecular structure of physalin A. (**B**) Breast cancer cell lines (MDA-MB-231, MDA-MB-453, and MCF-7 cells) were cultured with increasing concentration of physalin A (0, 10, 20, 40, 60, 80, 100 µM) for 24 h. The cytotoxic effect of physalin A was measured using the MTS assay. (**C**,**D**) Effect of physalin A on colony formation and migration of breast cancer cells. MDA-MB-231 and MCF-7 cells (2 × 10^3^ per well) were cultured with/without physalin A for 7 days. The colonies were scanned using a scanner. Migrations with/without physalin A were imaged at 0 and 12 h (scale bar: 1,000 µm). The percent inhibition of cell migration was calculated using untreated well as 100%. (**E**) Physalin A inhibits mammosphere-forming ability. MDA-MB-231 cells (1 × 10^4^ per well), MDA-MB-453 (2 × 10^4^ cells/well) and MCF-7 cells (4 × 10^4^ per well) were cultured in 6-well ultra-low attachment plates with/without physalin A. Representative mammospheres in the photos were captured by inverted light microscopy (scale bar: 100 µm). The mammosphere formation efficiency (MFE) was determined as shown in the graph. Mean ± SD values from three independent experiments are presented. * *p* < 0.005, ** *p* < 0.01.

**Figure 2 ijms-22-08718-f002:**
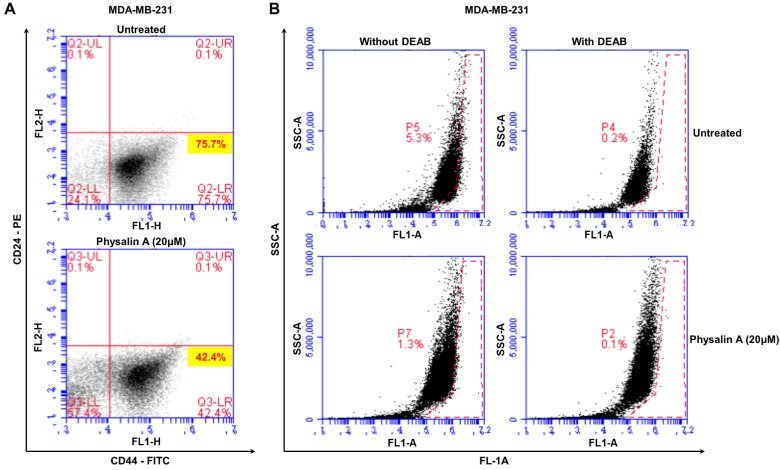
Effects of physalin A in CD44^high^/CD24^low^ and ALDH1-expressing breast cancer cells. CD44^high^/CD24^low^-cell subpopulation was measured by flow cytometry after treatment with physalin A (20 µM). (**A**) Effects of physalin A on CD44^high^/CD24^low^-expressing breast cancer cells. 1 × 10^6^ cells were cultured for 1 day and then treated with physalin A. After 1 day, cells were incubated with FITC-CD44 and PE-CD24 (BD, San Diego, CA, USA) and placed on ice for 20 min. Breast cancer cells were washed twice with 1X FACS buffer and assayed using flow cytometry (Accuri C6). The red cross was based on the binding of an antibody without physalin A. (**B**) MDA-MB-231 breast cancer cells were also treated with physalin A for 1 day. ALDH assay was performed using the ALDEFLUOR kit (StemCell Technologies). Breast cancer cells were incubated in ALDH assay buffer at 37 °C for 20 min. ALDH1-positive cells were determined using flow cytometry (Accuri C6).

**Figure 3 ijms-22-08718-f003:**
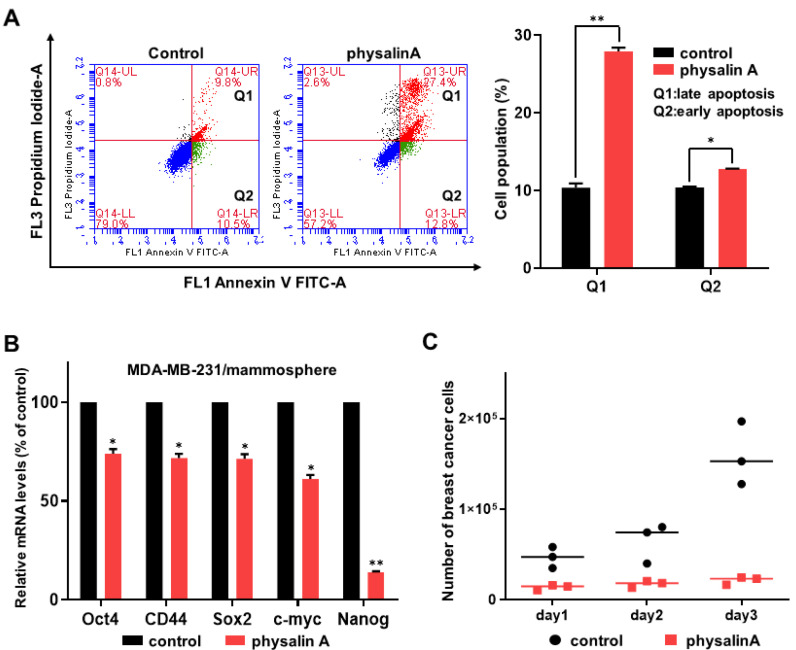
Effects of physalin A on apoptosis of mammospheres, expression of cancer stem cell markers, and mammosphere growth. (**A**) Physalin A increased apoptotic subpopulation of mammospheres. Mammospheres were cultured in an ultralow attachment 6-well plate for 5 days and then treated with/without physalin A for 2 days. Subsequently, the cells were collected and trypsinized to obtain single cells and washed with 1X PBS. 1 × 10^5^ single cells were counted and suspended with 100 µL of 1X Annexin V binding buffer. 5 µL of FITC Annexin V solution and 5 µL of PI staining solution were added to each cell. All samples were incubated for 15 min at room temperature. After washing with 1X Annexin V binding buffer, cell pellets were suspended with the buffer. Apoptosis was assayed by Annexin V/PI staining using an Accuri C6 flow cytometer (BD, San Jose, CA, USA). (**B**) Transcriptional levels of CSC markers, such as *Oct4*, *CD44, Sox2, c-myc*, and *Nanog* genes were determined in mammospheres treated with/without physalin A using CSC marker-specific primers and real-time qPCR (Appendix A). β-actin was used as an internal control. (**C**) Mammosphere growth is decreased by physalin A. MDA-MB-231 mammospheres treated with/without physalin A were suspended with a single cell, and the single cells were cultured in equal numbers in 6 cm diameter dishes. One to three days later, the cells were counted. Mean ± SD values from three independent experiments are presented. * *p* < 0.05, ** *p* < 0.01 vs. DMSO-treated control.

**Figure 4 ijms-22-08718-f004:**
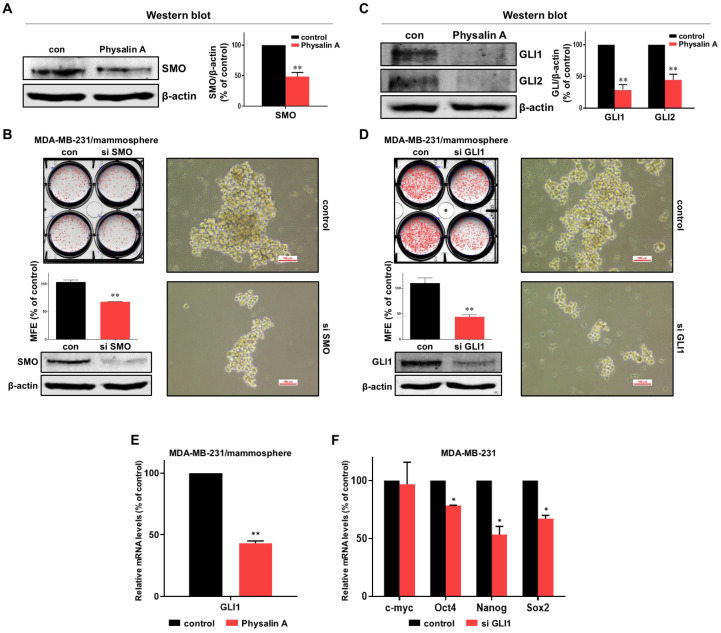
Effect of physalin A on Hedgehog signaling pathway. (**A**) Physalin A decreased the total protein level of SMO in breast mammospheres. An Anti-SMO antibody was used for immunoblotting. (**B**) Effect of SMO on mammosphere formation using SMO small interfering RNA (siRNA). siRNA-transfected cancer cells were incubated for 5 days with a complete Mammocults medium, and mammosphere formation efficiency (MFE) was calculated. (**C**) The protein levels of GLI1 and GLI2 in physalin A-treated mammospheres were assessed using specific anti-GLI1 and anti-GLI2 antibodies. (**D**) The effect of GLI1 protein on mammosphere formation was assessed using siRNA of GLI1. GLI1 siRNA-transfected cells were cultured for 7 days in a complete MammoCult medium and the MFE was determined. (**E**) Physalin A treatment reduced Hedgehog signaling-related *GLI1* gene expression in the mammospheres derived from MDA-MB-231 cells. (**F**) After transfection of GLI1 siRNA, total RNA was extracted from breast cancer cells. The transcripts levels of *c-myc, Oct4, Nanog,* and *Sox2* were measured by RT-qPCR using specific primers. β-actin was used as the internal control. Mean ± SD values from three independent experiments are presented. * *p* < 0.05, ** *p* < 0.01 vs. control.

**Figure 5 ijms-22-08718-f005:**
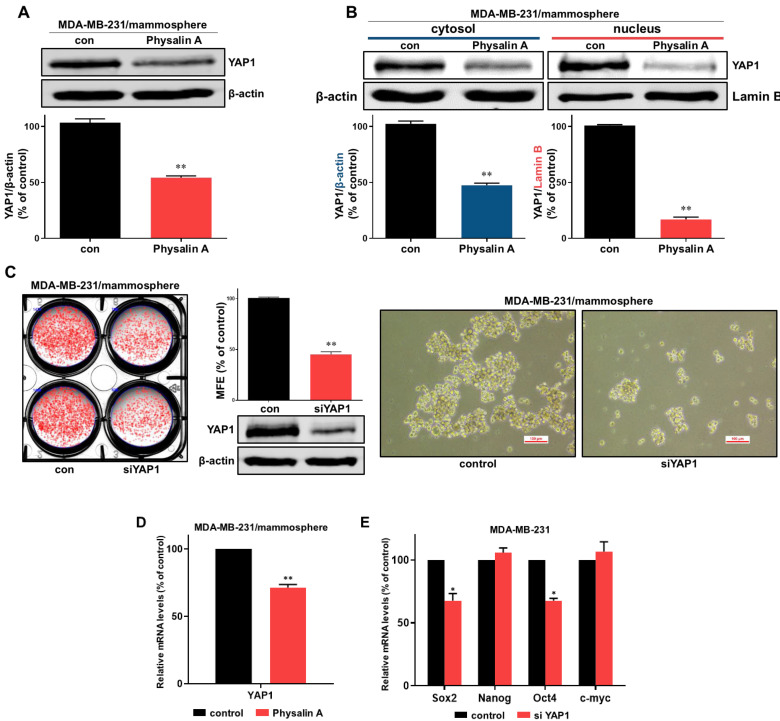
Effect of physalin A on Hippo signaling pathway. (**A**) Physalin A decreased the total protein level of YAP1 in breast mammospheres. Anti-YAP1 antibody was used for immunoblotting. (**B**) Treatment of MDA-MB-231 mammospheres with physalin A decreased the cytosolic and nuclear protein level of YAP1. (**C**) Effect of YAP1 on mammosphere formation using YAP1 small interfering RNA (siRNA). siRNA-transfected cancer cells were incubated for 5 days with a complete Mammocults medium, and mammosphere formation efficiency (MFE) was calculated. (**D**) Physalin A treatment reduced Hippo signaling-related *YAP1* gene expression in the mammospheres derived from MDA-MB-231 cells. (**E**) After transfection of YAP1 siRNA, the total RNA was extracted from breast cancer cells. The transcript levels of *c-myc, Oct4, Nanog*, and *Sox2* were measured by RT-qPCR using specific primers. β-actin was used as the internal control. Representative Western blot images of triplicate experiments are shown as mean ± SD. * *p* < 0.05, ** *p* < 0.01 vs. DMSO-treated control.

**Figure 6 ijms-22-08718-f006:**
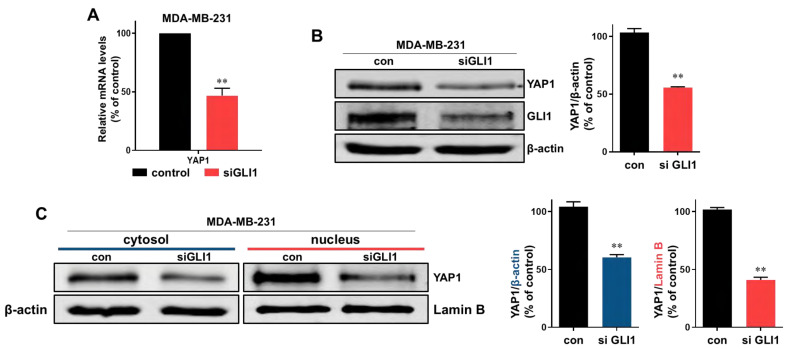
Down-regulation of YAP1 by knockdown of GLI1. (**A**) Effect of down-regulation of GLI1 on *YAP1* gene expression. MDA-MB-231 cells were transfected with siRNA of GLI1. After extraction of protein, the transcript level of YAP1 was assessed. (**B**) Effect of down-regulation of GLI1 on YAP1 protein expression. MDA-MB-231 cells were transfected with siRNA of GLI1. After extraction of protein, the protein levels of YAP1 were assessed. (**C**) The cytosolic and nuclear protein levels of YAP1 were also decreased in cells transfected with siRNA of GLI1. Representative Western blot images of triplicate experiments are shown as mean ± SD. ** *p* < 0.01 vs. DMSO-treated control.

**Figure 7 ijms-22-08718-f007:**
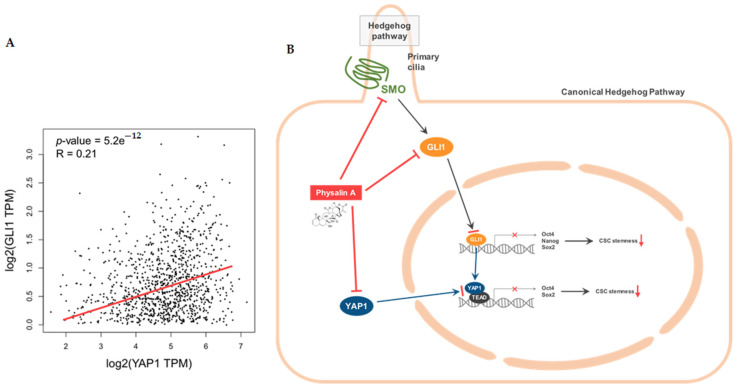
Relationship between GLI1 and YAP1 and the proposed model for physalin A-induced CSC death. (**A**) Scatter-plot of GLI1 and YAP1 expression in breast cancer patients (based on publicly available TCGA data). Spearman’s correlation coefficient and *p*-values are shown in this analysis. (**B**) The proposed schema suggests that physalin A regulates the canonical Hedgehog signaling pathway, induces down-regulation of YAP1, and inhibits BCSC. GLI1 also regulates the expression of YAP1. Inhibition of GLI1 and YAP1 contributes to the inhibition of BCSC formation.

## Data Availability

The data presented in this study are available within the article. Other data that support the findings of this study are available upon request from the corresponding authors.

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
