# Peer review of "Physalin A, 13,14-Seco-16, 24-Cyclo-Steroid, Inhibits Stemness of Breast Cancer Cells by Regulation of Hedgehog Signaling Pathway and Yes-Associated Protein 1 (YAP1)"

_ijms, 2021, doi:10.3390/ijms22168718_

Round 1

Reviewer 1 Report

ijms-1281677 Review

Physalin A, 13,14-seco-16, 24-cyclo-steroid, Inhibits Stemness of Breast Cancer Cells by Regulation of Hedgehog Signaling Pathway and Yes-associated Protein 1 (YAP1)

This article is written well, but lacks in new knowledge and grounds. Therefore, I require Major Revision.

Major point

・Please clarify the purpose of this study in “Introduction” section.

・Please describe the analysis method in this research in more detail.

・To prove Stemness in this study, you must show involvement of the several stem cell markers.

・The Western blotting figure is unclear and difficult to understand (Figure 2, Figure 4B,D). Please provide a clearer picture.

・Additional experiments are necessary to prove this theme. You should provide proof with a clinical specimen (Future clinical application is expected).

・In this research, verification is also necessary for intrinsic subtypes as subjects.

・In this study, prove it in breast cancer cell lines of each intrinsic subtypes.

・Please show results a little more clearly.

・Please indicate future clinical applications in the "Discussion section".

Minor point

・The sentence of this paper has many careful mention errors. Please review it.

Author Response

We submit answers of reviewer comments.

Reviewer 2 Report

The authors studied stem cell ness of breast cancer and the effects of physalin A through Hedgehog pathways and YAP1 in breast cancer. It is interesting topic, however, they need to improve several points.

In the introduction, the authors summarized only about triple negative breast cancer, but I think they need to discuss overall breast cancer. Cancer stem cells are not important only for triple negative breast cancer.

In this paper, the authors used two cell lines as a model of breast cancer. MDA MB 231 can be a model of triple negative breast cancer, however, MCF 7 has both estrogen and progesterone receptors, but not HER2 receptors. The authors should use triple negative breast cancer cell lines.

In Figure 1, the authors showed both MDA MB 231 and MCF 7, however, after that the authors focus on only one cell line. I think they cannot conclude the effects of physalin A by only one cell line.

Figure 4D, the band for the control is not clear.

Figure 5B, in the different figures, Physalin A is spelled out, but not in Figure 5B.  

I appreciate the authors tried to show their real experiment results, however the dots on the western can be removed by mixing blocking powders well.

Discussion

`Breast cancer is the most malignant cancer in women [32].`

Breast cancer is the most common malignant cancer in women. If they want to talk about triple negative breast cancer, they should write their features more. 

‘Hedgehog signaling pathway effectors, SMO and GLI, are essential for cancer progression. In a mouse model of pancreatic cancer, inhibition of Hedgehog signaling was shown to contribute to enhanced delivery of chemotherapy [42].’

These sentences did not make sense. Delivery of chemotherapy and cancer progression are not related.

I think the authors should re-write Discussion.

Author Response

we submit answers of review comments.

Round 2

Reviewer 1 Report

ijms-1281677 Review

Physalin A, 13,14-seco-16, 24-cyclo-steroid, Inhibits Stemness of Breast Cancer Cells by Regulation of Hedgehog Signaling Pathway and Yes-associated Protein 1 (YAP1)

This paper has been improved according to instructions. I consider it acceptable.

Author Response

Thank you for your good comments.

Reviewer 2 Report

I don't think the authors did not respond correctly what I suggested. MCF 7 is NOT a triple negative cell line. They added another cell line as MDA MB 453, but this cell line is known as melanoma origin. Except Figure 1. they used only one cell line and tried to conclude the characteristics of Physalin A - I think they need to choose another breast cancer cell line and need further study. I also suggested they should not focus on only triple negative breast cancer.

Author Response

We submitted answer of reviewer 2 comments.
